# Dynamic Responses of Major Equity Markets to the US Fear Index

**Bahram Adrangi** [1,*], **Arjun Chatrath** [2], **Joseph Macri** [3] **and Kambiz Raffiee** [4]

[1] W.E. Nelson Professor of Financial Economics, University of Portland, 5000 N. Willamette Blvd., Portland, OR 97203, USA

[2] Shulte Professor of Finance, Pamplin School of Business, University of Portland, 5000 N. Willamette Blvd., Portland, OR 97203, USA; chatrath@up.edu

[3] Department of Economics Macquarie, University North Ryde, Sydney, NSW 2109, Australia; joseph.macri@mq.edu.au

[4] Foundation Professor of Economics, College of Business, University of Nevada, Reno, NV 89557, USA; raffiee@unr.edu

\* Correspondence: adrangi@up.edu

**Abstract:** This study examines the reaction of four major equity markets of the world to the US equity market fear index, i.e., the Chicago Board of Trade Volatility Index (VIX). The VIX is designed to perform as a leading indicator of the volatility in equity markets. Our paper examines the daily data for the period of 2013 through 2018. We find that during this period there were three significant breaks in the data. Impulse responses from the structural vector autoregressive model estimation show that, in the first and second subperiods that cover from 6/2013 through 5/2016, equity market volatility in the US, UK, France, and Germany responded to structural shocks to the VIX. Nonlinear Granger causality tests confirm these findings. However, in the post Brexit-vote era, equity indices neither react to VIX structural shocks nor are caused by these shocks.

**Keywords:** volatility; international equity markets; structural vector autoregression; GARCH models; causality

**JEL Classification:** G10; G15; G17

---

## 1. Introduction

This paper examines the response of international equity markets to the S&P 500 implied volatility, the Chicago Board of Trade Volatility Index (VIX). It is well established that the VIX is, typically, negatively related to the performance of the market, being especially high when the market turns sharply negative. Therefore, the VIX is often regarded as the fear index within the US. The extent to which it also correlates to other major markets has been researched to a lesser extent. Moreover, the distortions arising from potential structural breaks in the VIX–market performance relationship have not been well attended to in the literature. This paper aims to provide evidence to this end.

Financial professionals and academics in the field have long experimented with developing ways of measuring volatility in the financial markets (e.g., Mills and Markellos 2008). Probably the most important of these measures is implied volatility—the implied standard deviation, traditionally obtained from working back from transaction prices of options via the Black Scholes framework. The Chicago Board Options Exchange (CBOE) introduced the first volatility index, which was known as VXO, in 1993. It was based on implied volatilities from at-the-money options on the SP100 index using a methodology proposed by Whaley (1993). The CBOE used an alternative methodology in 2003 to calculate VIX as a weighted sum of out-of-money option prices for all S&P 500 strikes in real time.

Carr and Wu (2006) show that the new VIX traces the volatility swap rate. Whaley (2008) discusses the public and media interest in the value of VIX as a measure of volatility. He explains the origin and purpose of creating VIX and its role in explaining the state of the economy and equity markets.

The VIX employs the near-month expirations on S&P 500 options and hence is associated with "expected" volatility over the coming month. In fact, many studies have shown the VIX to be the best (though not always perfect) predictor of future realized volatility amongst the different volatility frameworks (e.g., ARCH type) that also are thought to provide forecasting power. As such, implied volatility (in particular, the VIX) is found to be reasonably reliable indicator of future volatility, despite have some upward bias and periods of poor forecasting performance.[1] Moreover, there is also a strong association between VIX and contemporaneous price dynamics—positive shocks to VIX are associated with declining markets, and vice versa. It follows, therefore, that an elevated VIX portends weaker prices in the future.

There have been other approaches to assessing market sentiments and fear. Investopedia developed an Anxiety Index (IAI) that is also designed to gauge investor sentiment based on the actions of 30 million Investopedia readers globally. Unlike the VIX, it tracks investors' and readers' interest in 12 financial terms. The qualitative data based on these terms are extracted from more than 100,000 URLs dating back to pre-2008. The search of URLs gleans patterns and the usage of a dozen terms that may be related to investor fear, like "default," and opportunistic terms, like "short-selling." The percentage change in the measure of interest in these terms constitute the IAI over time. The IAI and the VIX are remarkably similar in terms of measuring market jitters. Figure 1 shows that the two indices move in tandem confirming that, while computed from different sources and by different methodologies, they reflect almost identical information regarding investor sentiment. Therefore, VIX as the "fear index" has substantial and quite direct empirical support.[2] On the other hand, a comparison of Figures 1 and 2 shows that movements in S&P 500 have not always been consistent with predictions of the VIX. Despite its flaws, the VIX remains arguably the most important gauge of market fear in the US.

This study examines the reactions of four major equity indices of the US, UK, Germany, and France to shocks to the VIX. Figure 3 shows the VIX with major breaks. The objective is to determine whether investors are sensitive to movements in the VIX. If it is found that the VIX performs as a leading indicator of equity market movements, it may provide useful information for the valuation of implied volatility-based derivatives, hedging strategies based on VIX fluctuations, and option pricing, among others.

---

[1] Academic and the popular sources have argued that the VIX may suffer from computational flaws and perhaps even manipulated. For instance, there has been periods that VIX predictions and market movements have diverged during some time periods. As an example, for four months between 8 August 2017 and 8 November 2017, the VIX was up 19%, signaling rising fear among market participants. This would imply a downward trend in S&P 500. However, the S&P 500 was rising in an upward trajectory. Other periods of strong divergences between VIX and S&P 500 are April 2007 to October 2007 and December 2014 to February 2015.

[2] Other implied volatility indices have been devised following the CBOE VIX, including: the VXN and VXD in the CBOE, the VDAX-NEW in Germany, the VX1 and VX6 in France, and the VSTOXX in the Eurex, among others.

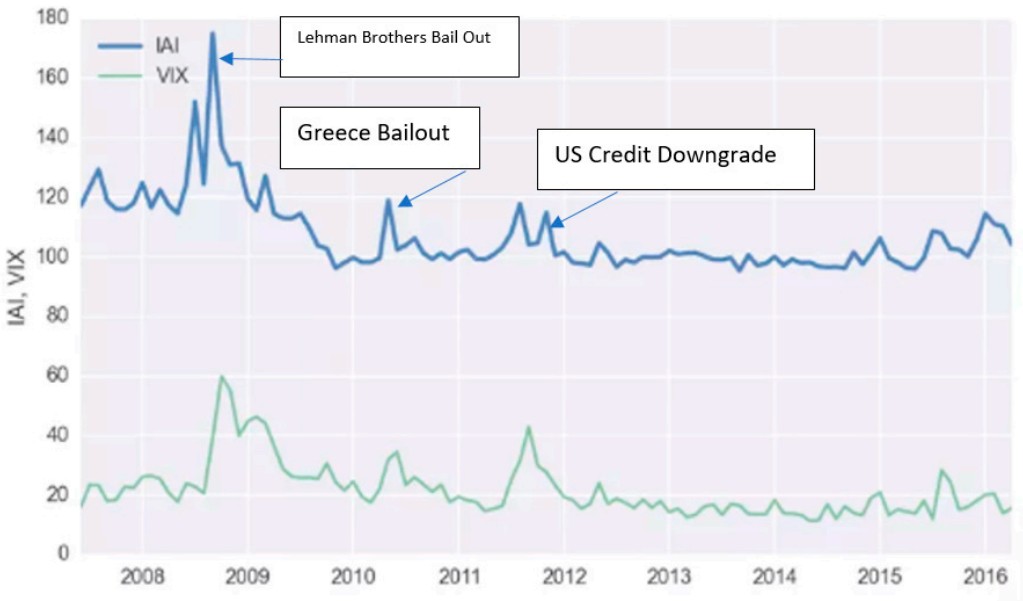

**Figure 1.** Anxiety and VIX Index. Source: www.investopedia.com/anxiety-index-explained.

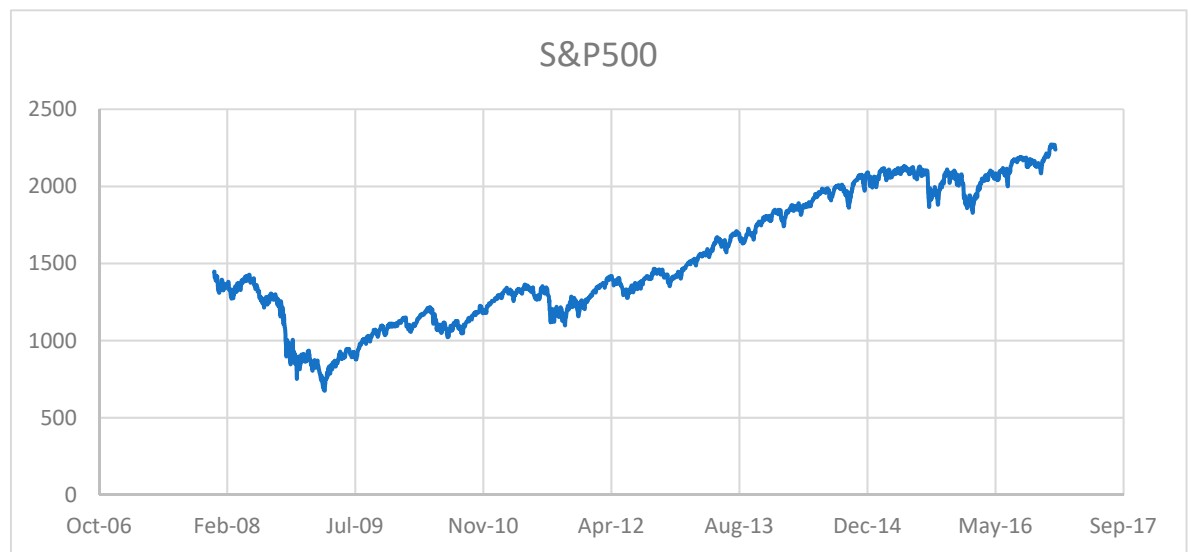

**Figure 2.** S&P500 INDEX. Source: Standard and Poor's.

The methodology of this investigation differs from many prior studies. Previous researchers (see Sarwar (2012a, 2012b) and Fleming et al. (1995)) deploy multivariate regression with change in the VIX as the dependent variable. There are several problems with this approach. First, the issue of stationarity of VIX and equity indices is glanced over perhaps because authors are using changes in VIX and returns on indices, which typically would be stationary. However, nonstationary series may be cointegrated, and it is well established by Johansen and Juselius (1990) and others that non-stationary series may be cointegrated. If so, first differencing and other transformations may result in the loss of valuable information about the long-term equilibrium relation among variables. Error correction models would be the most appropriate empirical approach in these situations.

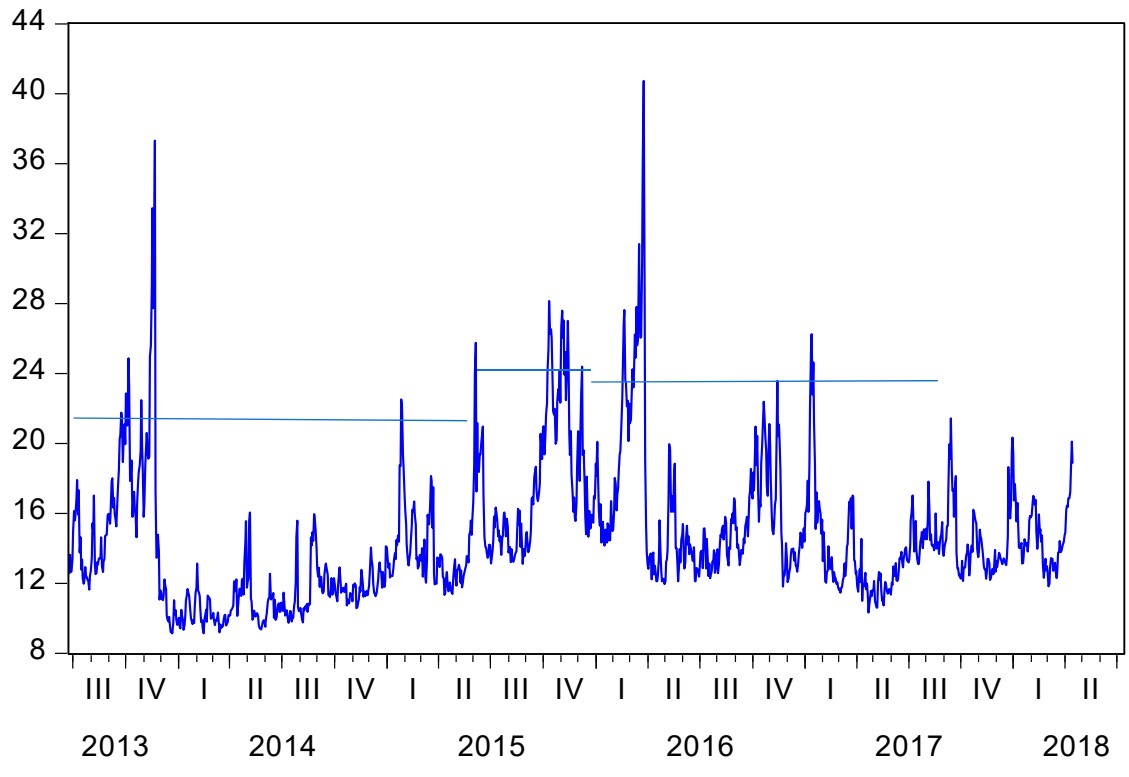

**Figure 3.** VIX and major breaks.

Second, practitioners view the VIX and similar fear indices as a predictor of market movements in the near future. That is the main reason that the VIX was created by CBOE as a leading indicator. Placing the VIX as the dependent variable and regressing it on index returns may be testing whether variations in the VIX are accounted for by equity index returns. However, the VIX, by its construction, is based on option prices on the S&P500. Option prices are related to spot movements of the S&P 500. Therefore, it seems highly plausible that the VIX is a function of movements in S&P 500, albeit, through options on S&P 500. Thus, regressing VIX on S&P 500 returns may be suffering from serious endogeneity problem. Third, it is not surprising that all previous research found a negative relationship between changes in VIX and S&P 500 returns. As Whaley (2008) explains, the VIX is dominated by put options. Therefore, as equity prices, and thus, the equity indices fluctuate, put option premiums move in the opposite direction. Therefore, the VIX is expected to be negatively associated with equity index movements. Thus, findings of almost all the research papers that estimate multivariate regressions with changes in the VIX as the dependent variable merely confirm the construction of the VIX. These results do not confirm whether the VIX is a leading indicator of equity market movements.

Finally, as evidence in the market shows, the relationship between the VIX and S&P 500 is not consistently negative. There have been many periods that VIX predictions and market movements have diverged, inconsistent with predictions of the VIX. In other words, there may be structural breaks in the implied volatility–market performance relationship. As indicated earlier, there have been many instances when the VIX and S&P 500 relationship diverged from their historic negative relationship.

These observations suggest that a fresh look and alternative modeling approach may be necessary using a different methodology. Our study formally tests for structural breaks in the VIX for the entire period of the sample. We identify three distinct subperiods and estimate a SVAR for each one. In each subperiod, we test the response of individual equity market indices under study to structural shocks to the VIX. This approach allows us to examine the distinct reaction by equities in each market to VIX shocks. Finally, detecting nonlinearities in all the time series under consideration, we deploy the nonlinear Granger causality tests (see Skalin and Teräsvirta (1999)) to examine the robustness of our findings regarding the predictive power of the VIX as a leading indicator of market movements

globally. These findings collectively may have implications for hedging strategies, especially if findings suggest disparate reaction to VIX movement in major equity markets.

The remainder of the paper is organized as follows. Section 2 offers a brief review of the relevant literature. Data are explained in Section 3. Methodology of the paper is the subject of the fourth section. Section 5 consists of the analysis of the empirical findings. The last section is devoted to the summary and conclusions.

## 2. Review of the Relevant Literature

Many studies have investigated the association between US equity market and the VIX (Fleming et al. (1995); Whaley (2000, 2009); Connolly et al. (2005); Giot (2005), among others). These studies almost unanimously find a substantial negative contemporaneous association between the VIX and US equity returns. In a related study, Durand et al. (2011) show that the market risk and value premiums in the Fama and French (1993) three-factor model respond to changes in the VIX.

Several papers in the last decade have researched the association between the VIX and other asset classes. Notable among them are papers that investigate fixed income securities, commodities, and foreign currencies, among others (See Badshah et al. (2013); Boscaljan and Clark (2013); Jubinski and Lipton (2013); Sari et al. (2011)).

Other studies extend this investigation to cross market associations of the VIX and equity markets of Europe, BRICS, Latin American equity markets, and other emerging markets. Notable among this research are Sarwar (2012a, 2012b) and Sarwar and Khan (2017), among others. Studies that consider implied volatility indices and their properties in other markets include Skiadopoulos (2004) in Frijns et al. (2010), who investigate similar relationships in the Australian equity market.

Many of these studies unequivocally confirm the negative relation between equity market returns and the VIX. These findings are not surprising as co-movement among equity markets of the US and rest of the world has been well-documented (Rapach et al. (2013) and Yunus (2013)). It is reasonable to expect that the negative association of the VIX with US equities will spill into other markets that show co-movement with the US equity markets.

The following is a brief summary of some of recent research on the VIX. We classify these papers into four groups. We discuss papers that examine the time series properties, information content, predictive power, and cross market applications of the VIX.

*2.1. Time Series Properties*

Psychoyios et al. (2010) set out to examine properties of the VIX data. They show that the logarithm of the VIX time series is stationary that includes jumps. They also demonstrate the effects of volatility on the VIX and the dynamics of the VIX during the periods of market upheaval. Their research suggests models for valuation and pricing of the VIX spot and forward options.

Fernandes et al. (2014) examine the time series properties of the daily VIX index from 2 January 1992 to 10 December 2008. Their main objective is to examine the value of the VIX for trading strategies. They confirm findings that there are long-term linear dependencies in the VIX. Their various statistical methodologies confirm and negative linear association between the VIX and S&P 500 index returns. The VIX and S&P 500 volume are also showing a linear positive contemporaneous association according to their empirical findings. They find some other interesting information on the VIX. They show a negative association between the VIX index and long-run movements in the oil price. The VIX shows no long-term association with the term spread despite a transitory positive contemporaneous positive relationship. Finally, the VIX index is not associated with the deviation in the Fed rates or on the credit spread, and only weakly related to the exchange value of the dollar. Their findings further establish that a semiparametric heterogeneous autoregressive (HAR) processes explains the VIX time series and may be used for forecasting the VIX.

Dotsis et al. (2007) analyzed continuous time diffusion and jump diffusion processes. Their findings show that the models that are based on random jumps may be the most consistent with major

volatility indices. This finding may cast some doubt on the predictive power of volatility indices. Wagner and Szimayer (2004) also corroborate these findings and show that the implied volatility indices VIX and VDAX demonstrate significant positive jumps.

Time series dynamics of the autonomous implied volatility process and its modeling has attracted the attention of scholars. Bakshi et al. (2006) estimated various general specifications of diffusion processes with a non-linear drift and diffusion component. The authors considered the squared implied volatility index VIX as a proxy to the unobserved instantaneous variance.

## 2.2. Information Content of the VIX

Several studies have researched the properties of implied volatility indices (e.g., Fleming et al. (1995); Moraux et al. (1999); Whaley (2000); Blair et al. (2001, 2010); Corrado and Miller (2005); Simon (2003) and Giot (2005)). The findings show that, while indices of implied volatility provide a fairly practical forecast of observed future market volatility, they are not free of error.

Jiang and Tian (2007) study the information content of the VIX. Their simulations reveal that under normal market conditions the VIX may under- or overestimate the true volatility by as much as 1.98 percent or overestimate it by as much as 0.79 percent. Their findings suggest that these biases could result in significant financial ramifications for investors. They propose an alternative measure based on smooth interpolation-extrapolation of the implied volatility function.

Becker et al. (2009) research the information content of the VIX. They show that the VIX is based on implied volatility, which is market-determined. It benefits from incorporating information derived from the historic jumps and expected future jumps in the market. This construct may be superior to econometric models that are strictly based on past volatility data.

## 2.3. Cross Market Association between the VIX and Equities

Notable among papers that investigate association of the VIX with equity indices and across markets of the world are by Sarwar and Khan (2017), Smales (2016), and Sarwar (2012a, 2012b), who apply the Fleming et al. (1995) model for testing VIX interactions with BRIC countries in the period 1993–2007. Results underline that the VIX is a fear gauge for China and Brazil during the whole period and for India during the subperiod 1993–1997. A similar investigation is carried out for six European countries in the 1998–2013 timeframe (Sarwar 2014), confirming the role of the VIX as a cross-market fear measure.

Smales (2016) examines the relationship between the VIX and multiple US, Australian, and New Zealand investment returns. His regression results using daily data for the period of 2001–2015, support a statistically significant negative association between the VIX and almost all assets under study with the exception of the US dollar. Specifically, rises in the VIX are associated with decline in stock markets, bond yields, and high-yielding currencies (AUD and NZD). The relationship is particularly prominent during the financial crisis of 2008–2009. His regression results also place om value on the VIX as a predictor of asset returns. In sum, his conclusion is that movements in the VIX may be useful in developing trading strategies and confirm findings of academic research that show close association between the VIX and financial market returns. Analysis of returns following periods of extreme investor fear may indicate some forecasting power for future returns, and this may therefore be used for devising profitable trading strategy. These findings confirm that investor fear and market returns are closely related.

Sarwar (2012b) employs regression analysis and daily data spanning 1993–2017 for the US and BRIC equity markets. His empirical findings suggest a strong negative contemporaneous association between the changes in the VIX and US and most BRICS stock market returns. The negative relationship in all markets is much stronger when the VIX is at high levels and volatile.

However, equity market responses to the VIX are asymmetric in reaction to positive change in VIX versus drops in the fear index. They conclude that the VIX is a gauge of investor fear across equity markets of the US and BRICS.

Sarwar and Khan (2017) examine the daily data on the VIX and equity indices of Lain America from 2003 through 2014. Their regression findings confirm that, in periods before, during, and after the financial crisis of 2008–2009, changes in the VIX and equity returns of major Latin American economies demonstrate a statistically significant and negative contemporaneous and delayed association. Their GARCH model shows that heightened levels of the VIX and its increased volatility is associated with depressed equity returns and higher volatility in the markets under study.

Sarwar (2012a) studies the relationship between the VIX and returns of the S&P 100, 500, and 600 indexes for the period 1992–2011 employing a dynamic regression model. His study takes structural breaks and divides the data into three subperiods of 1992–1997, 1998–2011, and 2004–2011 into account. His empirical findings support a significant negative contemporaneous association between daily changes in VIX and S&P 100, 500, and 600 returns. Furthermore, the level of volatility in the VIX proves to be a significant factor in magnifying the negative relationship between the VIX and index returns. Results also indicate a robust asymmetric association between stock market returns and changes in the VIX for all subperiods. The VIX is more sensitive to negative changes in equity returns than positive ones. This confirms that the VIX is a better gauge of investor fear than investor optimism.

Neffelli and Resta (2018) follow other researchers and examine the relationships between the VIX volatility and the US and BRIC market indices for the period of 2007 to 2018. Their empirical evidence confirms findings of (Fleming et al. (1995)) and (Sarwar (2012a, 2012b)). They also find that the role of the VIX for the US is stronger than reported by others for their time period. Their findings also support findings of Sarwar (2012b) and others when BRICS indices are considered. Specifically, they show that the VIX reflects investor fear in Brazilian IBOV, the Chinese SHSEC, the Indian BSESN, and the Russian IMOEX. In all markets under study, the role of the VIX was more pronounced during the financial crisis of 2008, as would be expected.

Ang et al. (2006) show that stocks that are sensitive to volatility shocks suffer low average returns as do those with high idiosyncratic volatility. Empirical evidence shows that low average returns are not accounted for by variables such as size, book-to-market value, and liquidity effects, among others.

## 2.4. The Predictive Power of the VIX

Another group of papers examine the role of the VIX in predicting volatility in equity markets. Blair et al. (2010) evaluate the information derived from the implied volatilities and intraday returns in forecasting index volatility over horizons from 1 to 20 days. Compared to ex post volatility, forecasts indicate that the VIX index provides useful information for this purpose.

Bekaert and Hoerova (2014) investigate the predictive power of the VIX. They decompose the squared VIX index into the conditional variance of stock returns and the equity variance premium. Their empirical findings show that the two components are reliable predictors of the economic activity and equity returns, respectively.

Giot (2005) finds a negative and statistically significant association between the returns of the S&P100 and NASDAQ100 indices and VIX and VXN indices. The relationship between the VIX and these equity indices is asymmetric, as negative and positive index returns are associated with larger and smaller changes in the VIX. The relationship between indices and the changes in the VIX also vary during the periods of high and low volatility. The VXN reaction to the indices under study is not especially pronounced. Turning to the question of predictability of future movements in indices, they show some evidence that positive (negative) future returns may follow extremely high (low) levels of the implied volatility indices.

Corrado and Miller (2005) investigate the forecasting power of the various implied volatility indices, including the VIX. They conclude that the forecast quality of VXO and the VIX has improved since 1995. Implied volatilities for the Nasdaq 100, i.e., VXN is shown to lead to even more accurate forecasts of future market volatility.

## 3. Data

The daily data for the study for the period covers 24 June 2013 through 12 July 2018. The daily index values for CAC 40 (CAC), DAX (Frankfort), FTSE (London), S&P 500 (US), and the VIX are taken from the Bloomberg data base. These indices represent the main national pan-European stock exchange group.

The CAC40 is one of the main indices in pan-European stock exchange group. It is a capitalization-weighted measure of the 40 most companies among the 100 highest market capitalization on the Euronext Paris. Similarly, DAX 30 (DAX) is an index of the performance of 30 largest blue-chip companies trading on the Frankfort Stock Exchange. Similar to CAC and the Dow Jones index, DAX does not necessarily reflect the movements of the broader market.

FTSE is the Financial Times Stock Exchange 100 Index. The index is based on the 100 highest market capitalization companies listed on the London Stock Exchange. Standard and Poor's 500 index known as S&P. The S&P 500 is also based on the market capitalizations of 500 large companies listed on the NYSE (New York), NASDAQ (over the counter), or the CBOE BZX Exchange.

The S&P 500 index is chosen to represent the US equity market, which constitutes 40 percent of the total market capitalization of the world equity markets. FTSE and DAX represent major European equity markets, constituting roughly 25 and 13 percent of the total market capitalization in Europe. CAC40 represents roughly 24 percent of the entire European market capitalization through Euronext Paris.

The CBOE Volatility Index (VIX) measures the expected price fluctuations in the S&P 500 Index options over the next 30 days. The VIX is known as the "fear index" and is calculated in real time by the CBOE. The VIX and its predecessor, VOX, have been indices used by the market participants to gauge the market sentiment in the US and around the world. We use the VIX as an indicator of future financial market risk because financial press quote the VIX volatility index as an investor fear gauge. Governmental agencies and central banks use the VIX as a barometer to assess risk in equity markets.

## 4. Methodology

*Structural Vector Autoregressive Formulation*

The main tool for the empirical investigation in this paper is a Structural Vector Autoregression model (SVAR). The methodology is well-developed in the literature. Following Adrangi et al. (2018), we formulate the following SVAR:

$$\mathbf{A}\,\mathbf{X_t} = \mathbf{B}_0 + \sum_{i=1}^{s} \mathbf{B_i}\mathbf{X_{t-i}} + \mathbf{u_t}, \tag{1}$$

where **A** is a $n \times n$ square matrix, in our case $5 \times 5$ because we have five endogenous variables.

$$\mathbf{A} = \begin{pmatrix} 1 & a_{12} & a_{13} & a_{14} & a_{15} \\ a_{21} & 1 & a_{23} & a_{24} & a_{25} \\ a_{31} & a_{32} & 1 & a_{34} & a_{25} \\ a_{41} & a_{42} & a_{43} & 1 & a_{45} \\ a_{51} & a_{52} & a_{53} & a_{54} & 1 \end{pmatrix}, \mathbf{u_t} = (\, u_t^{VIX},\, u_t^{cac},\, u_t^{DAX},\, u_t^{FTS},\, u_t^{SP}\,)',$$

where matrix **A** represents the structural model coefficients, vector $\mathbf{u_t}$ comprises of structural shocks, and vector $\mathbf{X_t} = (VIX_t, GARCAC_t, GARDAX_t, GARFTS_t, GARSP_t)'$ are the model variables. The GAR prefix stands for the time varying variances from a GARCH (1,1) model.

The off diagonal elements of matrix **A** represent the contemporaneous relationship among the five by one ($5 \times 1$) elements of the vector of the model stationary endogenous variables, i.e., VIX and time varying volatility in the five indices. $\mathbf{B}_0$ is a $5 \times 1$ vector of intercepts. $\mathbf{B_i}$ is a $5 \times 5 \times s$ coefficient matrix of lagged endogenous variables on the right-hand-side of the Equation (1). There are $5 \times 5 \times s$

(s is the lag order) parameters to be estimated in the matrix $\mathbf{B_i}$. We determine the lag order of the endogenous variables in SVAR be based on statistical criteria while estimating the model. The vector of white noise structural innovations (shocks) is the $5 \times 1$ vector $\mathbf{u_t}$ with elements that are uncorrelated with the model endogenous variables and across equations.

Multiplying both sides of the Equation (1) by the matrix $\mathbf{A}^{-1}$ produces the reduced form of the VAR, i.e.,

$$\mathbf{X_t} = \mathbf{G}_0 + \sum_{i=1}^{s} \mathbf{G_i X_{t-i}} + \mathbf{e_t}, \tag{2}$$

where, $\mathbf{G}_0 = \mathbf{A}^{-1} \times \mathbf{B}_0$, $\mathbf{G_i} = \mathbf{A}^{-1} \times \mathbf{B}$ and $\mathbf{e_t} = \mathbf{A}^{-1} \times \mathbf{u_t}$. The elements of vector $\mathbf{e_t}$, i.e., forecast errors, are a linear function of the structural innovations given by vector $\mathbf{u_t}$.

In order to derive the structural coefficients, initially the reduced form SVAR is estimated, the structural parameters are subsequently recovered to examine the responses of the model variables to structural shocks to each variable. Equation (3) shows that the structural shocks are a linear combination of the forecast errors in the reduced form expressed in Equation (2).

$$\begin{bmatrix} u_t^{VIX} \\ u_t^{CAC} \\ u_t^{DAX} \\ u_t^{FTS} \\ u_t^{SP} \end{bmatrix} = \begin{pmatrix} 1 & a_{12} & a_{13} & a_{14} & a_{15} \\ a_{21} & 1 & a_{23} & a_{24} & a_{25} \\ a_{31} & a_{32} & 1 & a_{34} & a_{25} \\ a_{41} & a_{42} & a_{43} & 1 & a_{45} \\ a_{51} & a_{52} & a_{53} & a_{54} & 1 \end{pmatrix} \times \begin{bmatrix} e_t^{VIX} \\ e_t^{CAC} \\ e_t^{DAX} \\ e_t^{FTS} \\ e_t^{SP} \end{bmatrix} \tag{3}$$

In Equation (3), the elements of vector $\mathbf{e_t}$ are the forecast errors associated with the VIX, and the time varying volatility in CAC, DAX, FTS, and SP, in the reduced form of Equation (1), respectively.

SVAR estimation leads to deriving impulse responses and prediction error innovation accounting, i.e., variance decomposition. In order to obtain the impulse responses and perform the innovation accounting, the estimates of the reduced form coefficients and the covariance matrix of the forecast errors in the reduced form are needed to obtain the structural model coefficients and innovations. Using an identification strategy, the structural shocks in Equation (1), i.e., the elements of vector $\mathbf{u_t}$, are fully recoverable from the forecast errors in the reduced form model by Equation (2).

The identification problem in SVAR models arises because the number of estimated coefficients derived from the reduced form estimation are not sufficient to recover the coefficients of the structural model and structural shocks. Therefore, some restrictions on the off diagonal elements of the matrix $\mathbf{A}$ in Equation (1) are necessary.

For instance, given the five endogenous variables, we would need ten necessary restrictions, i.e., $(n^2 - n)/2$, imposed on the elements of matrix $\mathbf{A}$, where $n = 5$ in this study. These restrictions are sufficient to render the remaining unrestricted elements $a_i$ of matrix $\mathbf{A}$ in Equation (3) identifiable.

Researchers offer various identification strategies. For instance, Sims (1989); Sims et al. (1990) and Kilian and Park (2009), among others, suggest a recursive identification strategy. This strategy entails imposing plausible restrictions on the contemporaneous relationships among the model variables in vector $\mathbf{x}$.

We impose long-run restriction that accumulated impulse responses of equity markets to shocks to the VIX shocks are negligible (Blanchard and Quah (1989)). This assumption is plausible because, while the shocks to VIX may trigger volatility in equity markets, in the long-run economic fundamentals and firm cash flows determine the direction of equities and indices. The reduced form forecast errors are a function of structural errors as

$$\mathbf{\Psi e_t = F u}_t$$

where matrix $\mathbf{\Psi}$ is an inverse matrix with long-run multipliers as its elements. From this equation, the reduced form errors and their covariance matrix may be computed as $\mathbf{e_t = \Psi^{-1} F u}_t$ and $\mathbf{E(e_t e_t')=\Sigma_e}$, where $\mathbf{\Sigma_e = \Psi^{-1} F F' \Psi^{-1}}$. The long-run identifying restrictions are imposed by setting elements of

matrix **F** equal to zero. For instance, $F_{ij} = 0$ implies that the long-run accumulated impulse response of variable I to shocks to the variable j is zero.

Finally, the Wold representation of the estimated structural model in Equation (1) is written in an infinite moving average representation of the structural shocks as follows:

$$\mathbf{X}_t = \mathbf{\Omega} + \sum_{j=0}^{\infty} \mathbf{\Phi}_i \mathbf{u}_{t-j} \tag{4}$$

where **Ω** and **Φ** are the vector of intercepts and the matrix of infinite structural shocks, respectively.

As discussed by Adrangi et al. (2019), elements of matrix **Φ** in Equation (4) can be used to derive variable responses to structural shocks to other model variables. For instance, $\phi_{ij}(0)$ is the instantaneous impact of a shock to innovation *j* on endogenous variable *i* and is called the impact multiplier. One-period impact of shocks to innovations j on variable *i* in time period t are given by $\phi_{ij}(t)$. Furthermore, by performing innovation accounting, one can examine the forecast error variance or variance decomposition. If shocks to a structural innovation explain none of the forecast error variance of endogenous variable $x_j$, then the series $x_j$ is unrelated with the remaining endogenous variables of the model.

Imposing long-run restrictions require the variables in the estimated SVAR to be stationary. Table 1 shows that, in most subperiods, equity market indices under study are not stationary. Therefore, prior to estimating the SVAR, we derive the time varying volatility in each index by fitting GARCH (11) models to each index return.

$$R_t = \sum_{i=1}^{p} \pi_i R_{t-i} + \varepsilon_t. \tag{5}$$

where, $R_t$ represents percentage changes in each series. The lag length for each series is selected based on the Akaike (1974) criterion. The residual term ($\varepsilon_t$) represents the index movements that are purged of linear relationships and seasonal influences. The conditional variance equation of the model GARCH (1,1) model is given by Equation (6).

$$\sigma^2_{i,t} = \beta_i + \gamma_i u^2_{i,t-1} + \varphi_i \sigma^2_{i,t-1} \ i = 1 \text{ to } 4. \tag{6}$$

where $\sigma^2_{i,t}$ is the conditional variance, $u^2_{i,t-1}$ is the lagged innovations, and $\sigma^2_{i,t-1}$ is the lagged conditional volatility. The GARCH models are estimated by the Maximum likelihood method.

**Table 1.** Break points, diagnostics, and summary 24 June 2013–24 July 2018.

| Panel A: Bai Perron Test of Structural Breaks | | | |
|---|---|---|---|
| Break Test | Scaled F-Statistic | Critical Value * | Dates |
| 0 vs. 1 [b] | 43.310 | 11.47 | 8/12/2015 |
| 1 vs. 2 [b] | 20.905 | 12.95 | 5/17/2016 |
| 2 vs. 3 [b] | 19.799 | 14.03 | 2/13/2017 |
| 3 vs. 4 | 1.427 | 14.85 | |
| ADF Unit Root Test with Structural Break | | | |
| Based on Minimizing the Dickey-Fuller t-statistic 1st | | | −7.110 [a] |
| Based on Minimizing the Dickey-Fuller t-statistic 1st | | | −6.774 [a] |
| Based on Minimizing the Dickey-Fuller t-statistic 1st | | | −6.770 [a] |

**Table 1.** *Cont.*

| | | Panel B: Levels 6/24/2013–8/12/2015 | | | |
|---|---|---|---|---|---|
| Tests | CAC | DAX | FTSE | S&P | VIX |
| ADF | −2.826 | −2.409 | −4.216 [a] | −4.141 [a] | −5.058 [a] |
| PP | −2.847 | −2.409 | −4.392 [a] | −4.137 [a] | −4.753 [a] |
| KPPS | 0.284 [a] | 0.256 [a] | 0.109 | 0.390 [a] | 0.328 [a] |
| ARCH-LM | 6.934 [a] | 8.788 [a] | 7.848 [a] | 13.345 [a] | |

| | | Panel C: Levels 8/13/2015–5/17/2016 | | | |
|---|---|---|---|---|---|
| Tests | CAC | DAX | FTSE | S&P | VIX |
| ADF | −2.538 | −2.286 | −3.301 [c] | −2.394 | −3.052 |
| PP | −2.600 | 2.354 | −3.282 [c] | −2.442 | −2.911 |
| KPPS | 0.145 [b] | 0.152 [b] | 0.177 [b] | 0.165 [b] | 0.165 [b] |
| ARCH-LM | 6.934 [a] | 8.788 [a] | 7.848 [a] | 13.345 [a] | |

| | | Panel D: Levels 5/18/2016–2/13/2017 | | | |
|---|---|---|---|---|---|
| Tests | CAC | DAX | FTSE | S&P | VIX |
| ADF | −3.065 | −2.539 | −2.599 | −3.179 [c] | −4.024 [a] |
| PP | −3.192[c] | −2.616 | −2.581 | −3.166 [c] | −4.137 [a] |
| KPPS | 0.103 | 0.208 [b] | 0.081 | 0.200 [b] | 0.114 |
| ARCH-LM | 2.191 | 2.693[c] | 13.955 [a] | 29.097 [a] | |

| | | Panel E: Summary descriptive statistics for model variables. All variables are in level | | | |
|---|---|---|---|---|---|
| Statistics | CAC | DAX | FTSE | S&P | VIX |
| Mean | | 64.730 | 79733.720 | 107.271 | 72.388 |
| Stand Dev | | 8.231 | 27324.000 | 102.228 | 34.522 |
| Skewness | | 0.281 | 0.977 | 0.992 | 0.899 |
| Kurtosis | | 1.912 | 2.563 | 3.858 | 3.013 |
| J-B | | 33.149 [a] | 87.081 [a] | 103.658 | 71.473 [a] |

| Panel F: Johansen-Juselius Cointegration Test, unrestricted VAR lag order = 6 | | | | | |
|---|---|---|---|---|---|
| *r* = The number of cointegrating vectors among the four variables | | | | | |
| vector | | $\lambda_{m}$ | *p*-Value | $\lambda_{t}$ | *p*-Value |
| *r* = 0 | | 40.271 [a] | 0.001 | 70.191 [a] | 0.000 |
| *r* ≤ 1 | | 26.011 [a] | 0.009 | 29.919 [b] | 0.048 |
| *r* ≤ 2 | | 3.860 | 0.873 | 3.908 | 0.910 |
| *r* ≤ 3 | | 0.047 | 0.827 | 0.047 | 0.827 |

* Bai-Perron (Econometric Journal, 2003) critical values. Order of lags in the VAR for cointegration test is 6, determined by the AIC, SBC, FPE, and likelihood ratio test. KPSS tests include an intercept in the test regression. The null hypothesis in that the series is trend-stationary. Significance indicates nonstationary. Cointegration with unrestricted intercepts and no trends in the cointegrating VARs. *p*-values from MacKinnon et al. (1999) for both $\lambda_{m}$ and $\lambda_{t}$ reject no or one cointegrating vector. Maximum eigenvalue and traces tests suggest two cointegrating vectors at 5% level. [a], [b], and [c], represent significance at 0.01, 0.05, and 0.10, respectively.

## 5. Empirical Results

Before estimating the SVAR and examining the impulse responses, we plot the daily graphs of every time series under study.

The visual inspection of the time series plot of the equity indices show clear evidence that the indices are not stationary. These graphs are not provided for the purpose of brevity. The time series plot of the VIX reveals that, while the index may be stationary during certain subsets of the period under study, structural breaks are also visible. The focus of the paper is on shocks to the VIX and their impact on the equity indices under study. Therefore, we use the breaks in the VIX to determine the subperiod in which we examine the impulse responses.

The first panel of Table 1 shows the results of Bai-Perron test of structural breaks. The test statistic determines structural breaks determined by the time series. The panel A in Table 1 reports the results of this test.

The Bai-Perron test signals three structural breaks that occur on 12 August 2015, 17 May 2016, and 13 February 2017. The first break appears to coincide with several political and economic upheavals. The most important economic events may be the devaluation of yuan by China and the Greek sovereign debt crisis. Around this break, the European Union refused to extend credit relief to Greece, which threatened the financial stability of the banking system in Europe and elsewhere. On the political front, the Syrian refugee problem reached its peak, and the conflict in Syria intensified. In response, Russia entered the Syrian conflict.

Th second structural break appears to have been triggered by the Brexit discussions leading UK to the June 2016 referendum. The last break could have been due to the Brexit complications as well as threat of war in the Korean Peninsula triggered by the nuclear tests by North Korea. Therefore, we estimate three SVAR models, one for each subperiod.

Estimating the specified SVAR requires estimating an unrestricted VAR for stationary time series. Some variables of the SVAR model are nonstationary by the Augmented Dickey Fuller (ADF) and Phillips-Perron unit root tests, as well as (KSPSS) test of stationarity as shown in Table 1. Lütkepohl (2005), among others, suggest transforming SVAR variables to stationary for the SVAR to be stationary and stable. Non-stationary variables could result in a non-stationary VAR system and impulse responses that are spurious and do not die down with time.

Stock (1987), West (1988), and Sims et al. (1990) show that parameter estimates of VAR are consistent when variables are nonstationary, but small samples may result in biased estimates. Sims (1989) suggests that Bayesian estimation approach may be more appropriate for modeling and estimating VAR models with nonstationary variables. Given the difficulties with the choice of prior distribution for Bayesian estimation, we opt for estimating VARS with stationary variables.

A graphic examination of our time series variables show that all indices may be nonstationary. Stationarity and unit root tests reported in Table 1 confirm our visual conclusions.

Augmented Dickey Fuller (ADF) and Phillips-Perron (PP) tests for unit root, and the KPSS test of stationarity, show that most variables in every subperiod are non-stationary. S&P 500 and FTSE show some conflicting test results. For instance, in the first and the last subperiods, they are either stationary or border-line stationary. However, we are aiming to measure the volatility in each equity market. To this end, we introduce and estimate a GARCH (1,1) model for stationary percentage changes in each equity index in each subperiod. The objective is to extract the time-varying heteroscedasticity or volatility in each market and subperiod. We specify and estimate the GARCH model in Equations (5) and (6). ADF, PP, and KPSS tests of stationarity show that the variances over time are all stationary.

Unrestricted VARS for each subperiod are estimated with lag orders of eight. Multiple lag order criteria are employed because there may be conflicting signals by various criteria. For instance, Schwarz Baysian Criteria (SBC) tends to underestimate the number of lags. Too few lags could result in a non-stationary VAR system and residuals that are not white noise. Other lag order criteria like Hannon-Quinn (HQ) the Akaike Information Criterion (AIC), the Forecast Prediction Error (FPE), as well as the likelihood ratio test (LR) are also examined.

The VAR lag order for all subperiods by the Final Prediction Error (FPE), AIC, SC and HQ is determined to be one. The inverse roots of the characteristic polynomial are all inside of the unit circle confirming that the estimated VAR systems are stationary in the subperiods. This ensures that the impulse responses eventually die down. Imposing short-run restriction that matrix $\mathbf{A}^{-1}$ is lower triangular, we are able to derive the structural innovations vector $\mathbf{u}$ and the impulse responses. The impulse response function is the time path of the volatility in the four equity markets following a positive shock to the VIX index. Impulse responses show the size of the impact of a shock as well as the rate at which the response tapers off. The point estimates and their two-standard error bands are shown by the solid and dotted lines in all cases. Figure 4 presents these results.

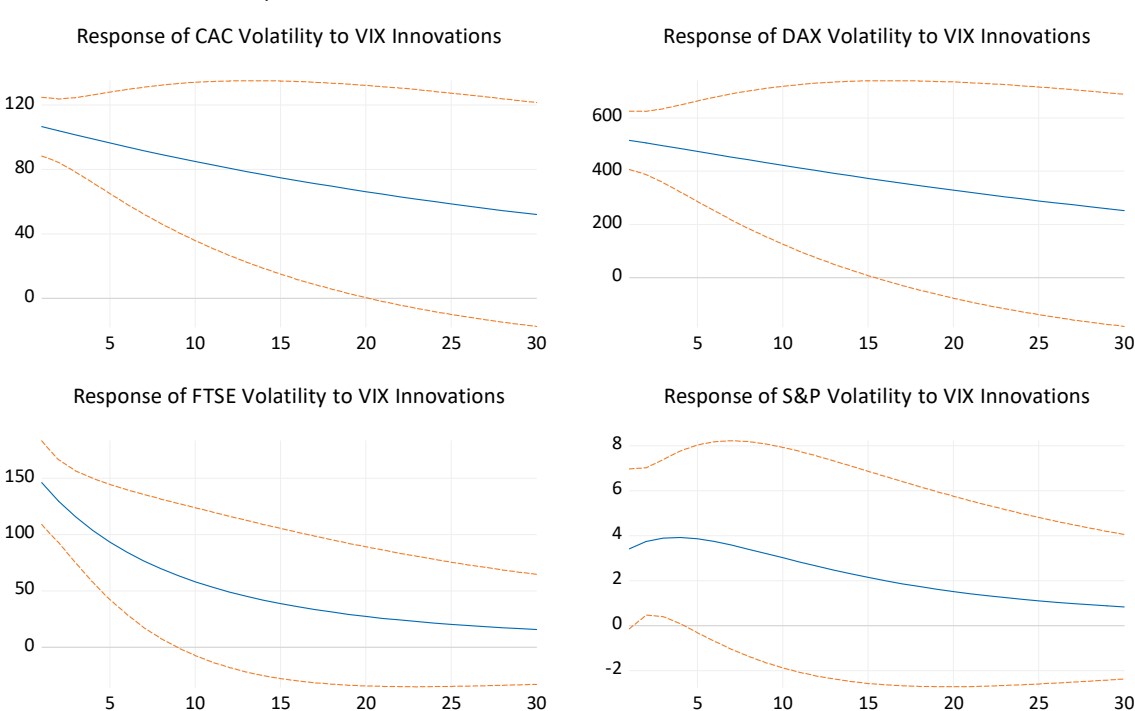

**Figure 4.** Responses of volatility in equity indices to structural innovations in the Chicago Board of Trade Volatility Index (VIX). Estimates and two standard deviation confidence band are in solid and dotted lines. SVAR lag order is one. Long-run identification restrictions are imposed. 24 June 2013–12 August 2015.

It is evident that a heightened VIX may lead to investor anxiety in all equity markets under consideration. The two standard deviation confidence interval indicates that the impulse responses are statistically significant. Therefore, rising fear reflected by positive shocks to the VIX results in investor anxiety in the four equity indices. The heightened volatility in all markets decline over the next 15 to 20 days and become statistically insignificant. The conclusion is that the VIX shocks do trigger investor fear and market volatility in the first sup-period under consideration.

One may conclude that the investing community in these major markets might have considered some of the events in this subperiod troubling. For instance, rejecting the bailout of Greek banks and demanding austerity and fiscal responsibility by Greece was a position that was supported in Europe but could have raised fears of another financial crisis. Also, Russia entering the Syrian war was widely viewed by investors as a sign of further escalation in the geopolitical landscape.

Accumulated impulse responses in the four equity markets to shocks to the VIX are presented in Figure 5. It is evident that there is a considerable accumulated volatility in response to shocks to the VIX index.

Table 2 presents the decomposition of the forecast error of equity indices explained by the SVAR variables in the first subperiod. For instance, the one day ahead forecast error variations of volatility in CAC is mostly due to the shocks to the CAC itself (53.16%), while 23.06% of the variance is accounted for by the VIX and 17.57% by the DAX. The share of FTSE and S&P in CAC volatility forecast error are negligible. As is shown in Table 2, in 30 days, the variations in CAC are mostly accounted for by other shocks to the CAC (72%), the VIX (15%), and the DAX (11%). We conclude that based on the variance decomposition, in the first subperiod, equity market events specific to France and French economy, the VIX, and the German equity market play a relatively significant role in the CAC volatility. This is plausible as German economy and sentiments in its equity market play a vital role in the European economies. The role of the VIX may be showing the sensitivity in CAC in response to the US market volatility. In summary, the volatility in equity markets of Europe show some response to the VIX.

In 30 days, the variations in CAC, DAX, and FTSE due to VIX ranges from 15% to 9%, respectively. One concludes that, while the European equity markets are sensitive to the US fear index, the main portion of volatility in these markets are due to co-movement among these markets.

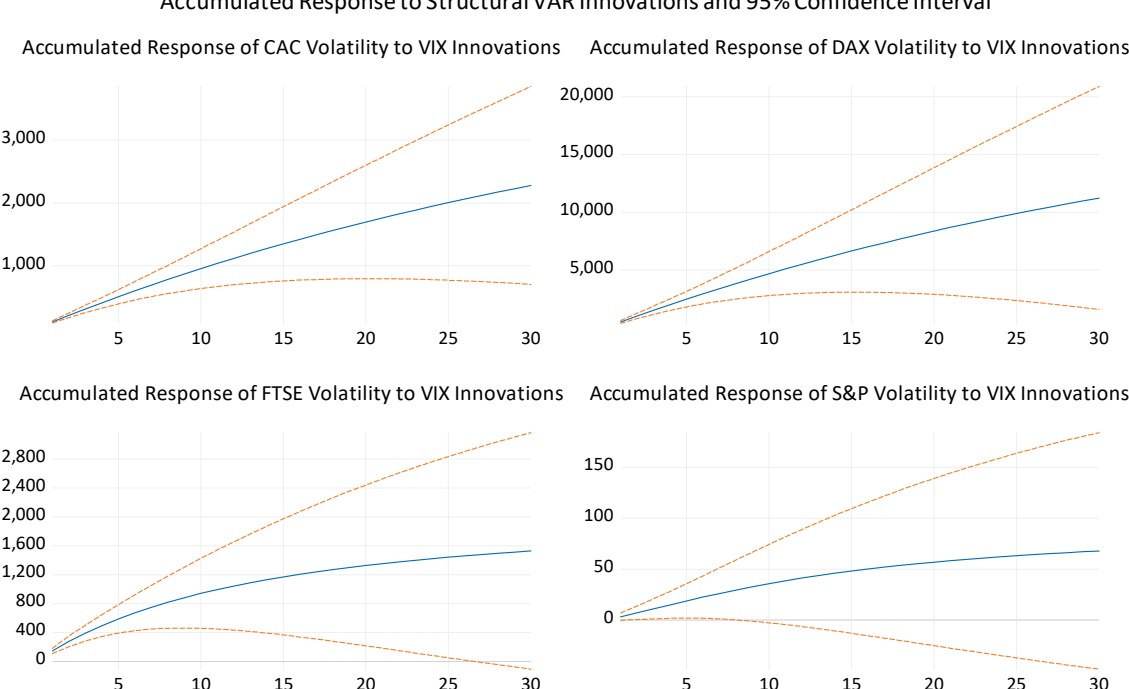

**Figure 5.** Accumulated responses of volatility in equity indices to structural innovations in VIX. Estimates and two standard deviation confidence band are in solid and dotted lines. SVAR lag order is one. Long-run identification restrictions are imposed. 24 June 2013–12 August 2015.

Impulse responses of volatilities in four equity indices derived from the moving average representation of the structural model for the second subperiod are presented in Figure 6. The two standard deviations bands are also provided. Impulse responses show that all equity markets experience rising volatility in response to structural shocks to the VIX, which level off in about 10 to 15 days. FTSE and S&P 500 initially show an unexpected decline in volatility, but the volatility rises after a few days in both markets. The two standard deviation confidence interval indicates the impulse responses are statistically significant up to around fifteen days at most.

The conclusion is that the VIX shocks trigger investor fear and market volatility in this sup-period. The increased volatility in CAC and DAX are the most short-lived. It is plausible that, with the Brexit anxieties in place, shocks to the VIX stirred a considerable degree of fear in European markets. The destabilizing effects of Brexit in the second subperiod were similar to some of the stabilizing effects of events in the first subperiod. Accumulated impulse responses in the four equity markets to shocks to the VIX are presented in Figure 7. It is evident that there is substantial and statistically significant accumulated volatility in response to shocks to the VIX. The accumulated impulse responses are consistent with the impulse reposes, which were on the negative side but were increasing. This observation bolsters the notion that Brexit might have exacerbated the impact of shocks to the VIX.

**Table 2.** Percentage of equity index volatility forecast error variations explained by VIX and Equity Indices, 6/24/2013–8/12/2015.

| CAC Period | S.E. | VIX | CAC | DAX | FTSE | S&P 500 |
|---|---|---|---|---|---|---|
| 1 | 1.523907 | 23.06727 | 53.16470 | 17.57172 | 6.186519 | 0.009798 |
| 10 | 3.325619 | 19.33011 | 62.15958 | 15.28370 | 3.158607 | 0.068011 |
| 20 | 3.553793 | 16.78376 | 68.39097 | 12.85680 | 1.914194 | 0.054273 |
| 30 | 3.596813 | 15.23153 | 72.33904 | 10.97169 | 1.416521 | 0.041231 |
| **DAX Period** | **S.E.** | **VIX** | **CAC** | **DAX** | **FTSE** | **S&P 500** |
| 1 | 222.0597 | 15.51648 | 74.49980 | 0.168141 | 9.667806 | 0.147774 |
| 10 | 688.4700 | 12.91648 | 82.06945 | 0.217239 | 4.644559 | 0.152269 |
| 20 | 935.9210 | 10.97281 | 85.79171 | 0.446709 | 2.689165 | 0.099600 |
| 30 | 1090.002 | 9.727636 | 87.40137 | 0.853079 | 1.945311 | 0.072604 |
| **FTSE Period** | **S.E.** | **VIX** | **CAC** | **DAX** | **FTSE** | **S&P 500** |
| 1 | 1308.442 | 11.12727 | 40.32318 | 7.713062 | 33.81403 | 7.022454 |
| 10 | 4132.530 | 9.762820 | 47.22903 | 8.129161 | 32.40228 | 2.476710 |
| 20 | 5707.291 | 9.125213 | 49.48482 | 8.540962 | 30.72729 | 2.121724 |
| 30 | 6718.869 | 8.916086 | 50.41408 | 8.771329 | 29.85478 | 2.043726 |
| **SP Period** | **S.E.** | **VIX** | **CAC** | **DAX** | **FTSE** | **S&P 500** |
| 1 | 438.4022 | 0.698788 | 20.86801 | 0.167246 | 8.772745 | 69.49321 |
| 10 | 993.6533 | 2.033265 | 32.29965 | 2.183065 | 16.41988 | 47.06415 |
| 20 | 1107.649 | 2.374618 | 35.12433 | 3.529921 | 17.92936 | 41.04177 |
| 30 | 1141.226 | 2.445579 | 35.85994 | 4.117280 | 18.04220 | 39.53501 |
| | | | Factorization: Structural | | | |

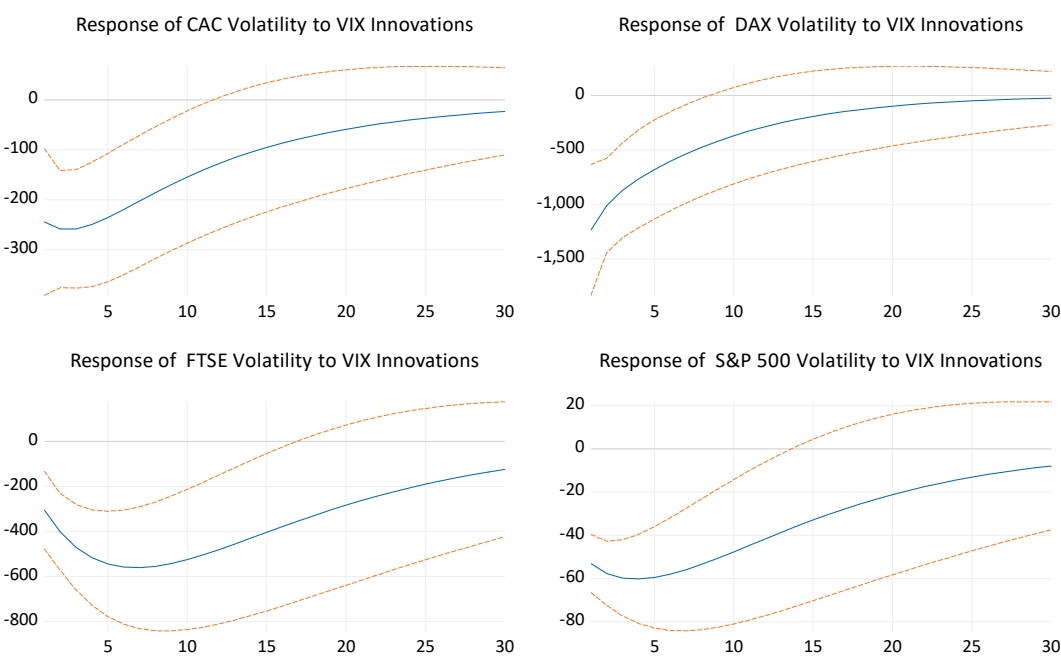

**Figure 6.** Responses of volatility in equity indices to structural innovations in VIX. Estimates and two standard deviation confidence band are in solid and dotted lines. SVAR lag order is one. 13 August 2015−17 May 2016.

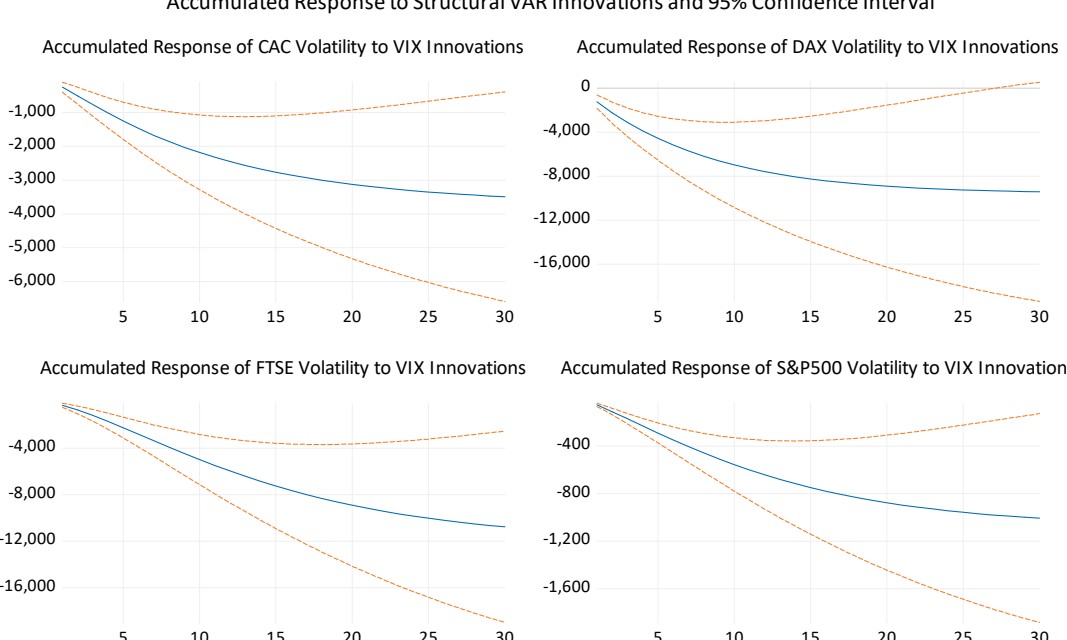

**Figure 7.** Responses of volatility in equity indices to structural innovations in VIX. Estimates and two standard deviation confidence band are in solid and dotted lines. SVAR lag order is one. 13 August 2015−17 May 2016.

Table 3 presents the decomposition of the forecast error of the equity indices explained by the SVAR variables in the second subperiod. A significant portion of one day ahead forecast error variations in the forecast error of volatility in all indices except S&P 500 are due to the shocks related S&P 500. For instance, S&P 500 shocks account for 41% to over 50% of one day ahead forecast error is CAC, DAX, and FTSE. In the case of S&P 500, a significant portion of its one day ahead error forecast is due to the VIX and CAC. Over 30 days, the share of error forecast accounted for by VIX in all markets rises. This could support the view that with Brexit afoot, all equity markets are jarred and the US equity market and the VIX are the source of movements and volatility in other markets. This may be supported by a significant portion of volatility in all markets being explained by CAC because the French economy and its equity market may be gaining a leading role. The role of the VIX may show the sensitivity in CAC in response to the US market volatility. In summary, the volatility in equity markets of Europe show significant response to VIX. In 30 days, the variations in CAC, DAX, FTSE, and S& P500 due to the VIX, range from 17% to 40%, respectively. One concludes that, in the period of post Brexit turmoil, the European equity markets are sensitive to the US fear index. A significant portion of volatility in these markets is also due to CAC, which signals a co-movement of European equity markets.

**Table 3.** Percentage of equity index volatility forecast error variations explained by VIX and Equity Indices, 13 August 2015–17 May 2016.

| CAC Period | S.E. | VIX | CAC | DAX | FTSE | S&P 500 |
|---|---|---|---|---|---|---|
| 1 | 1.847290 | 5.629366 | 42.89547 | 8.235468 | 1.730171 | 41.50953 |
| 10 | 4.213181 | 20.57815 | 46.04785 | 5.384713 | 1.144778 | 26.84451 |
| 20 | 4.803299 | 22.96667 | 44.30306 | 5.285415 | 1.125518 | 26.31934 |
| 30 | 5.022063 | 23.30053 | 44.05778 | 5.270821 | 1.125361 | 26.24551 |
| **DAX** | **S.E.** | **VIX** | **CAC** | **DAX** | **FTSE** | **S&P 500** |
| 1 | 1031.178 | 8.503843 | 40.11901 | 3.069205 | 0.485771 | 47.82217 |
| 10 | 1540.439 | 17.22094 | 45.11183 | 3.074743 | 0.384313 | 34.20818 |
| 20 | 1595.023 | 17.90871 | 44.26796 | 3.222857 | 0.390217 | 34.21025 |
| 30 | 1602.747 | 17.95395 | 44.20409 | 3.234434 | 0.390874 | 34.21664 |
| **FTSE** | **S.E.** | **VIX** | **CAC** | **DAX** | **FTSE** | **S&P 500** |
| 1 | 4240.179 | 6.352564 | 17.10188 | 17.15250 | 9.062495 | 50.33057 |
| 10 | 5679.630 | 31.55825 | 32.41167 | 7.387517 | 7.149734 | 21.49282 |
| 20 | 5779.511 | 38.40037 | 32.18627 | 5.762504 | 6.048016 | 17.60284 |
| 30 | 5786.676 | 39.48003 | 31.75040 | 5.582793 | 5.804940 | 17.38184 |
| **S&P 500** | **S.E.** | **VIX** | **CAC** | **DAX** | **FTSE** | **S&P 500** |
| 1 | 1212.531 | 27.93073 | 44.85573 | 5.416856 | 20.34589 | 1.450803 |
| 10 | 2840.446 | 38.13473 | 48.99117 | 1.206103 | 9.250807 | 2.417189 |
| 20 | 3281.835 | 39.88873 | 45.88734 | 1.480075 | 7.741952 | 5.001906 |
| 30 | 3377.665 | 40.11354 | 45.20379 | 1.576134 | 7.521768 | 5.584767 |

Factorization: Structural

Impulse responses of volatilities in four equity indices for the third subperiod are presented in Figure 8. The two standard deviations confidence interval indicates that, while they show statistically significant response to shocks to the VIX in all equity indices except DAX, they quickly turn statistically insignificant in about three days. The most significant possible reasons for shocks to the VIX during this period were uncertainty related to Brexit and the crisis over North Korean missile tests. We conclude that, by this time, the Brexit was discounted by all markets, and the North Korea affair was considered a passing issue that would not lead to a serious conflict. In the interest of brevity, we do not provide the accumulated impulse responses or the variance decomposition for the third subperiod.

As the last panel of Table 1 indicates, there is one cointegrating vector among the five time series in the study. Therefore, there is a long-run equilibrium relationship that captures the association of the time series under study. A vector error correction model (VECM) may be estimated that explains the short-run deviations from the equilibrium. Impulse responses from the VCEM for the first subperiod are presented in Figure 9. These impulse responses corroborate the observation from those of the SVAR. All equities exhibit a negative reaction to positive shocks to the VIX for this subperiod. Similarly, impulse responses for the remaining subperiods confirm those of the SVAR but are not presented here for the purpose of brevity.

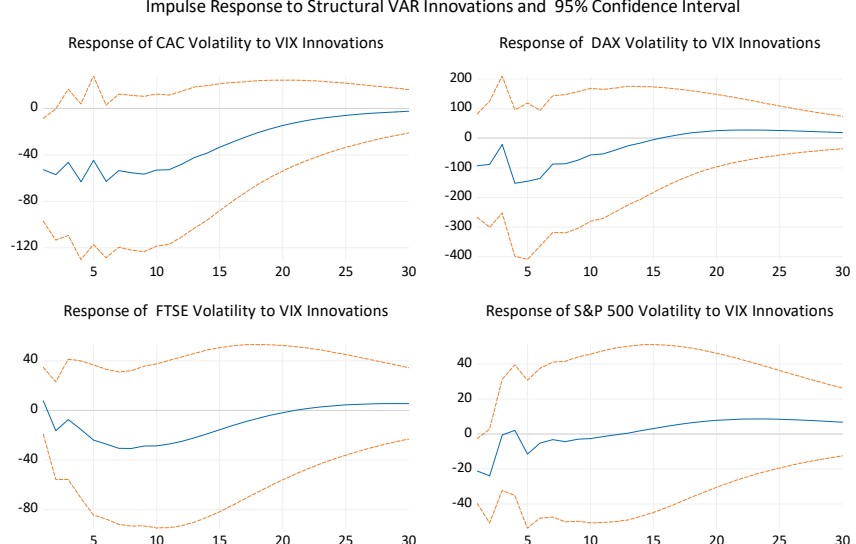

**Figure 8.** Responses of volatility in equity indices to structural innovations in VIX. Estimates and two standard deviation confidence band are in solid and dotted lines. SVAR lag order is one. 18 May 2016−13 February 2017.

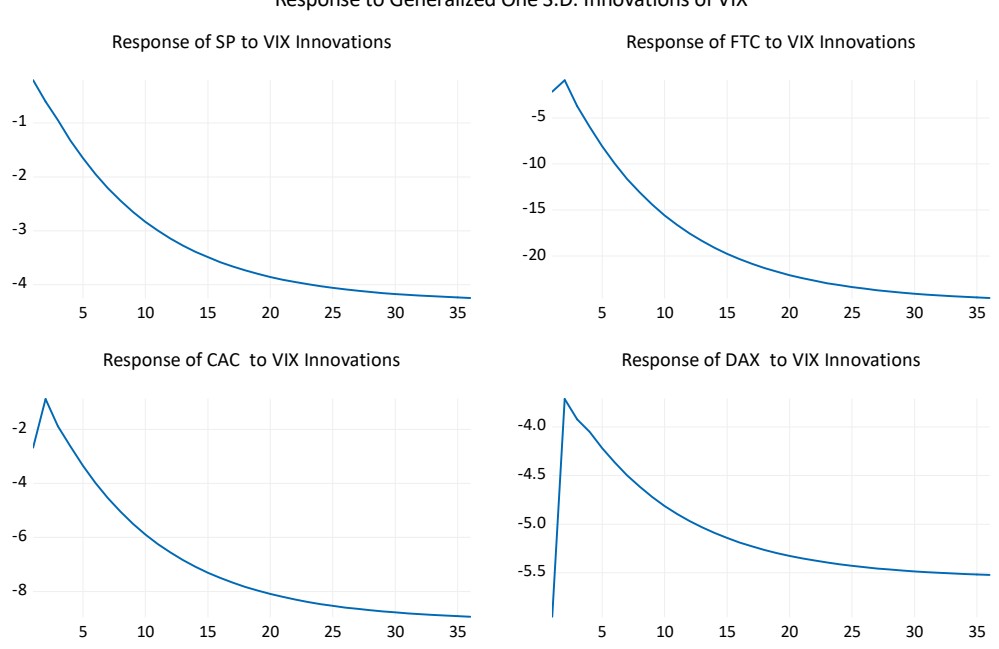

**Figure 9.** Responses of equity indices to innovations in the VIX from the vector error correction model. VECM lag order is one. 24 June 2013–12 August 2015.

*Nonlinear Causality Test Results*

We have established that the VIX triggers volatility in all equity markets in the first and second subperiods as indicated by impulse responses in market volatility and the variance decomposition. The responses to the VIX in all markets vary from subperiod to subperiod. We conclude that market responses to the VIX are not uniform and depend on a context based on other significant events. For instance, sensitivity to the VIX shocks for pre- and post-Brexit are quite different. Therefore, it may be informative to examine whether the relationship dynamics may be examined by testing for causality. However, after finding ARCH effects in all index series, there may be influences of complex nonlinearity in the series. Therefore, following Adrangi et al. (2015), we apply a nonlinear extension

to the standard Granger causality tests (Granger (1969), Geweke (1984)), which is based on smooth transition regression (STR). The non-linear impact of x on y is characterized by an additive smooth transition component. The following additive smooth transition regression model,

$$y_t = \pi_{10} + \pi_1'w_1 + (\pi_{20} + \pi_2 w_t)F(y_{t-d}) + \delta_1'v_t + (\delta_{20} + \delta'_2 u_1)G(x_{t-e}) + u_t \tag{7}$$

is specified, where $\delta_j = (\delta_{j1} \ldots \delta_{jq})'$, $j = 1, 2$, $v_t = (x_{t-1} \ldots x_{t-q})'$ and $G(.)$ is a transition function. Non-causality is tested as H0: $G \equiv 0$ & $\delta_{1i} = 0$, $i = 1 \ldots q$. The approximation to the above equation is

$$y_t = \overline{\pi}_{10} + \overline{\pi}_1'w_1 + (\pi_{20} + \pi_2 w_t)F(y_{t-d}) + k'v_t + \sum_{i=1}^{q}{}' \sum_{j=1}^{q} \phi_{ij} x_{t-1} x_{t-j} + \sum_{i=1}^{q} \psi_i x^3{}_{t-1} + u_t \tag{8}$$

where $K' = (k_1 \ldots k_q)$, and non-causality is supported by $k_i = 0$, $\varphi_{ij} = 0$ and $\psi_i = 0$, $i = 1 \ldots q$, $j = 1 \ldots q$. Under $H_0$, the resulting test statistic has an asymptotic $\chi^2$ distribution with $(q \times (q + 1)/2) + 2q$ degrees of freedom.

Table 4 presents the results of the nonlinear Granger causality tests for $q = 5 \ldots 10$. We report the P values for F statistics that test the joint null hypotheses of no causality, i.e., that $k_i = 0$, $\varphi_{ij} = 0$ and $\psi_i = 0$. Therefore, at some lag levels of variable x the null may not be rejected. Skalin and Teräsvirta (1999) vary the lag order to detect possible causality between variables at varying lags.

**Table 4.** Nonlinear Granger causality test: *p*-values of F statistics for the Ho of no nonlinear Granger causality.

| | | | Panel A: 24 June 2013–12 August 2015 | | |
|---|---|---|---|---|---|
| | Causing Variable | | | Caused Variables | |
| Lags | VIX | CAC | DAX | FTSE | S&P 500 |
| 5 | | 0.307 | 0.178 | 0.880 | 0.998 |
| 6 | | 0.002 | 0.012 | 0.361 | 0.086 |
| 7 | | 0.016 | 0.062 | 0.260 | 0.097 |
| 8 | | 0.027 | 0.083 | 0.124 | 0.958 |
| 9 | | 0.039 | 0.037 | 0.001 | 0.797 |
| 10 | | 0.033 | 0.016 | 0.004 | 0.901 |
| | | | Panel B: 13 August 2015–17 May 2016 | | |
| | | CAC | DAX | FTSE | S&P 500 |
| 5 | | 0.754 | 0.968 | 0.749 | 0.662 |
| 6 | | 0.618 | 0.882 | 0.007 | 0.048 |
| 7 | | 0.041 | 0.028 | 0.077 | 0.065 |
| 8 | | 0.074 | 0.094 | 0.073 | 0.003 |
| 9 | | 0.073 | 0.857 | 0.690 | 0.004 |
| 10 | | 0.706 | 0.707 | 0.474 | 0.536 |
| | | | Panel C: 18 May–13 February 2017 | | |
| | | CAC | DAX | FTSE | S&P 500 |
| 5 | | 0.996 | 0.999 | 0.701 | 0.101 |
| 6 | | 0.974 | 0.868 | 0.838 | 0.169 |
| 7 | | 0.984 | 0.943 | 0.806 | 0.317 |
| 8 | | 0.838 | 0.822 | 0.772 | 0.634 |
| 9 | | 0.561 | 0.647 | 0.608 | 0.499 |
| 10 | | 0.194 | 0.512 | 0.385 | 0.536 |

Notes: The reported *p*-values are for the F statistic for of the test for joint null hypotheses of no causality, i.e., that $k_i = 0$, $\varphi_{ij} = 0$ and $\psi_i = 0$. Therefore, at some lag levels of variable x the null may not be rejected. For instance, the computed *p*-values for the VIX causing FTSE for the 2015−2016 interval show that the former causes the latter for lags of six to eight days. The degrees of freedom in the numerator and the denominator of the F-test of causality are $q \times (q + 1)/2 + 2q$ and $T − n − q \times (q + 1)/2 − 2q$, respectively, where q is the number of lags, n is the dimension of the gradient vector, and T is the number of observations. Degrees of freedom in the numerator of the F statistics are 25, 32, 42, 52, 63, and 75 for $q = 5$ through 10 respectively.

For all cases and all lag levels, except for CAC and DAX in the first and second subperiods, many *p*-values of F statistics are less than 5%, showing that the null hypothesis of no causality can be rejected at a 5 and 10 percent levels in many cases for the first two subperiods. Thus, there is some evidence of causality from the VIX to major world equity markets, especially in the first two subperiods. Considering that causality is a stronger relationship than correlation and response to shocks, results are not surprising. The case for the causality between the VIX and world equity indices is supported, which corroborates the evidence shown by impulse responses of equity indices to VIX shocks as indicated by the impulse response functions.

It may be argued that, the for the first and second subperiods, the equity investors and markets were responsive to the geopolitical and economic events, but not during the third subperiod. For instance, by around February thirteenth, 2017, investor and markets might have been accustomed to the discussions around Brexit. Other research has shown that the saliency of issues in the minds of investors may shift over time. Furthermore, it is well known that investors and equity markets are sometime ready to respond to any uncertainty much more vigorously than other periods.

The reaction of the equity indices to VIX shocks is not uniform across subperiods. For instance, the French and German equity indices were much more sensitive to VIX shocks in the first subperiod. One main reason may be that the Syrian refugee crisis and other political events such as Russia's entry into the Syrian war were weighing heavily on the equity markets of the two countries. Both nations were in a political turmoil over the Syrian refugee influx into Europe. It became a politically divisive issue for both governments and ultimately lead to the weakening of the German coalition government. The uncertainty fallout undoubtedly jarred the investor confidence and equity markets in both economies. In the second subperiod, the causality between the VIX and S&P 500 is mostly statistically significant at less than 5% level. This could signal that the US investors and markets had grown sensitive to uncertainty and ready to react to minor shocks. This period signified a steady rise in S&P 500. Investors do become sensitive to market shocks as equities go on long upward trends.

## 6. Summary and Conclusions

The objective of this study is to examine the reaction of four major equity markets of the world to the US fear index, the VIX. Practitioners and academicians have been interested in the role of the VIX as a leading indicator of the volatility in equity markets. While the academic research presents a near unanimous negative relationship between the VIX and equity markets, practitioners observe many periods of divergence between the VIX and equity indices such as the S&P 500. The discord between the academic findings and market observations may be related to the regression methodologies that researchers have deployed.

Our paper examines the daily data for the period of 2013 through 2018. We find three significant breaks in the data during this period there were on the VIX according to Bai and Perron (2003) test. Breaks in data occur during significant economic and political upheavals such as the depreciation of the yuan, the Greek sovereign debt crisis, and Brexit.

We employ the structural vector autoregressive (SVAR) formulation that includes the VIX and volatility in the equity indices under consideration. Time-varying volatility in equity indices are derived from GARCH (1,1) model estimated for equity indices of US (S&P 500), UK (FTSE 100), France (CAC 30), and Germany (DAX 40).

Impulse responses from the SVAR estimation show that in the first and second subperiods that cover from June 2013 through May 2016, equity market volatility responds to structural shocks to the VIX. Nonlinear Granger causality tests confirm these findings. However, in the third subperiod that covers May 2016, through February 2017, and is characterized by geopolitical crisis in the Korean Peninsula as well as lingering complications surrounding Brexit referendum, the equity indices under study do not react to VIX structural shocks. Furthermore, we find no causal relationship between the VIX and equity indices in the third subperiod. Our results are in line with observations of practitioners and confirm that while the VIX may perform as a reliable leading indicator of market volatility during

some periods, this connection may be dynamic and sensitive to time periods under consideration. Market reaction to VIX shocks may depend on the saliency of issues dominating the sentiments of market participants. For instance, in the post Brexit referendum era, markets might have capitalized the negative information, and shocks to the VIX might have played a less significant role in the minds of players in the market. Therefore, the VIX may be treated as one indicator of the future equity market fluctuations in the major markets under study. However, shocks to the VIX may not trigger reaction in equity markets during some time periods.

**Author Contributions:** Data curation, B.A. and A.C.; Formal analysis, B.A., A.C., J.M. and K.R.; Methodology, B.A., A.C., J.M. and K.R.

**Funding:** This research received no external funding.

**Acknowledgments:** We are indebted to two anonymous reviewers for their constructive comments. Any remaining errors are our responsibility.

**Conflicts of Interest:** The authors declare no conflict of interest.

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
