# Peer review of "Dynamic Responses of Major Equity Markets to the US Fear Index"

_jrfm, doi:10.3390/jrfm12040156_

Round 1
Reviewer 1 Report
At first glance, the manuscript seems to be an intriguing contribution on a topic of interest in financial literature. The manuscript appears to be well-written and a consistent effort, that despite some merits suffers from a series of shortcomings.
Major concerns:
· I have serious doubts about the relevance of Figure 1 and especially Figure 2 that are placed in the Introduction.
· The introduction needs to be upgraded. The objective stipulated in lines 111-112 is quite straight forward, as the literature tends to highlight the impact of VIX on market sentiment. The section dealing with original contributions stretching from lines 115 to 155 is a welcomed sight, but I would recommend upgrades in terms of details, cohesion and consistency.
· The data section is the weakest part of the manuscript. I expect to see here a motivation on the choices of indices. Of course, they are relevant, but this is something of general knowledge. Another motivation should gravitate around opting for VIX out of the large battery of fear and greed and sentiment indexes brought forward by academics and practitioners.
· I will not comment on the methodology, as it appears to be again pretty straight-forward, and at first glance appears to be a logical choice.
· The result section is rich and extensive. It might benefit a little from some restructuring. Several graphical elements might be better suited in an Appendix section.
· The conclusions section is also not the strongest section. I would expect to see here a more in-depth formulation.
· Though industrious, well developed and logical, I strongly feel that the present approach would highly benefit from a/some robustness exercise/exercises. This would add weight to the results and general construction of the manuscript. The possibilities here are numerous and several strategies could be considered. This is the main reason for which I am not insisting on a specific option.
Minor concerns:
· Minor spelling errors throughout the entire article. See for example line 36 – GARCH (1,1) and not GARC (1,1). A language check-up would be a real bonus.
· Minor variations in the font used: see for example: line 495.
· Certain areas require a little rephrasing.
Author Response
Dear Reviewer:
Thank you for the review and the detailed comments. Please see the uploaded reply document. We hope that you find our replies satisfactory.

Reviewer 2 Report
This paper provides a very interesting analysis of the relationship of international equity markets to the US VIX index. The analysis is conducted in an SVAR framework and significant breaks are identified and linked to economic events (see section V).
The paper could be linked to other papers that have explored the role of the VIX internationally during previous crisis periods. Perhaps you could add some discussion of Lee et al (2014) on international relationships amongst VIX indices during the GFC. This paper is focused on the relationship amongst VIX indices and is a counterpart to your focus on the VIX and stock returns.
References
Lee Y.-H., Tucker A.L., Wang D.K., Pao H.-T. (2014), Global contagion of market sentiment during the US subprime crisis, Global Finance Journal, 25 (1) , pp. 17-26.
Author Response
Thank you for the careful review of our paper and the comments. In the attached document we respond to your comments. We hope you find the revised manuscript to your satisfaction. Thank you.

Round 2
Reviewer 1 Report
First of all, I want to congratulate the authors on their efforts in the direction of providing an updated version of the manuscript. I will use the same strategy as in the author reply and discuss each of the initial comments in the order of the original review.
The authors manage to partially answer some of the original comments. However, this does not alter my initial good opinion on the manuscript, despite the fact that I would have expected more work done in several areas.
Major concerns:
Comment 1. My initial comment aimed at the relevance of those figures as a part of the introduction, given the general norms of academic writing. However, given the ambiguity of my initial phrasing and the position of the authors, I am inclined to move past this observation.
Comment 2. I noticed only minor changes in the introduction along the lines specified in my previous comment. I would have expected far more done in this direction. Therefore, I can’t consider this action as passable.
Comment 3. I agree with the motivation of the authors.
Comment 5. I agree with the motivation of the authors.
Comment 6. I am very aware of what the current approach puts forward and my comment aimed above that point. I agree with the idea that implementing additional robustness would expand the size of the manuscript.
Minor concerns:
Comment 1. The problem has been solved.
Comment 2. The problem has been solved, hopefully. I can’t offer an explanation here on the technicalities of the submission engine.
Comment 3. The problem has been solved, at least to a certain extent.
Author Response
Thank you for the careful review of the manuscript. We are pleases that you find the majority of our responses to the first round of comments to your satisfaction. We hope that the introduction section of the revised manuscript will meet your expectations. Thank you again.
